# REVISITING THE LOTTERY TICKET HYPOTHESIS FOR PRE-TRAINED NETWORKS

## ABSTRACT

The lottery ticket hypothesis (LTH) suggests the possibility of pruning neural networks at initialization. Our study revisits LTH in the context of *transfer learning*, unveiling novel insights surpassing prior studies limited to LTH's application in pre-trained networks. To begin, our study shows that multiple pruning-at-initialization methods are likely to find worse pruning masks than a simple magnitude-based pruning method for pre-trained networks, owing to an inaccurate approximation of the influence of each weight. Iterative magnitude pruning (IMP) can find *trainable subnetworks* (*winning tickets*) even for pre-trained networks, however, IMP is a costly algorithm that requires multiple training cycles. Given that trainable subnetworks can be identified only when the initial network withstands the training's inherent randomness, and considering the superior resilience of pre-trained networks to this randomness compared to randomly initialized networks, we empirically demonstrate the enhanced efficiency of identifying trainable subnetworks within the framework of transfer learning. By challenging conventional wisdom surrounding *gradual magnitude pruning* (GMP), we reveal its capability to significantly enhance the trade-off between transfer learning performance and sparsity in terms of pruning-at-initialization. Our experiments, which involve various models such as convolutional neural networks and transformers, across both vision and language domains, demonstrate that GMP can identify trainable subnetworks for pre-trained networks at a significantly lower cost than IMP. For example, for ImageNet pre-trained ResNet-50, at a pruning ratio of 99%, GMP achieves comparable or superior results to IMP on the CIFAR, Caltech-101, Oxford-IIIT Pets, and Stanford Cars datasets, with 42 times less computation than IMP. Ultimately, we provide empirical evidence that the methodological distinction between the LTH-based and conventional pruning methods can be blurred for pre-trained networks.

## 1 INTRODUCTION

Transfer learning methods (Weiss et al., 2016) learn various features by solving pretext tasks on large datasets and then using pre-trained models as initialization to learn downstream tasks. Owing to the increase in model and dataset sizes and the development of pretext tasks, transfer learning has achieved excellent performance in many tasks; thus, it has become the de facto standard in computer vision and natural language processing. However, the large number of parameters in pre-trained models, designed to learn rich features, becomes problematic for end-users with limited memory and computation budgets when they aim to apply these pre-trained models to their downstream tasks. In a reality where not all creators of foundation models readily offer highly lightweight versions of their models, efficient training at the downstream level is essential to address the issue. In parallel with the increase in the size of deep learning models, studies on *pruning-at-initialization* (Frankle & Carbin, 2018; Lee et al., 2018) have attracted attention. For example, Frankle & Carbin (2018) established the lottery ticket hypothesis (LTH), which states that a sparse network that can achieve a performance similar to that of the full network with the same number of updates exists in a randomly initialized model. They proposed *iterative magnitude pruning* (IMP) that repeats train-prune cycles with weight resetting–resetting weights to initial values–to obtain a sparse trainable network of the target sparsity.

Most studies on pruning-at-initialization have been conducted from the perspective of model training from scratch (Frankle & Carbin, 2018; Lee et al., 2018). However, as transfer learning has become more prominent, we explore the LTH in the context of transfer learning. We explore the methodology

for identifying a trainable subnetwork within a pre-trained network *using only a downstream dataset*. This inquiry gains significance given the proliferation of large-scale pre-training datasets that remain undisclosed (Zhai et al., 2022). We first show that *saliency-based* pruning-at-initialization methods are suboptimal for pre-trained networks. Specifically, we prune an ImageNet (Deng et al., 2009) pre-trained ResNet (He et al., 2016) with SNIP (Lee et al., 2018), GraSP (Wang et al., 2020), SynFlow (Tanaka et al., 2020), and ProsPr (Alizadeh et al., 2022) using various downstream datasets; the results show that compared to the sparse network obtained by simply pruning the pre-trained model based on the magnitude of its weight values (Han et al., 2015b), the pruning-at-initialization methods lead to models that significantly underperform at all pruning ratios–the percentage of weights that are pruned–tested. We attribute such failures of the existing methods to the inaccurate approximation of the influence of each weight (referred to as "synaptic saliency") and empirically demonstrate that the synaptic saliency diverges from the true saliency as the pruning ratio increases. Despite the failings of the pruning-at-initialization methods, we observe that IMP successfully finds trainable subnetworks even for pre-trained networks. These observations are consistent with the outcomes of Chen et al. (2020), which validated the application of the LTH by employing IMP on pre-trained BERT networks (Devlin et al., 2018). However, IMP is a computationally intensive algorithm that requires multiple training cycles. Moreover, IMP requires significantly more train-prune cycles as the target sparsity increases because the increase in the pruning ratio between training cycles exponentially decreases.

In our study, we show that *IMP is not essential to identify trainable subnetworks in the context of transfer learning*. Our hypothesis can be derived from related studies (Frankle et al., 2020a; Paul et al., 2022) that define "stability" for an initial network and show the ease of finding trainable subnetworks for *stable* initial networks. After probing whether pre-trained networks are significantly more stable than randomly initialized networks, we apply various types of pruning algorithms (Han et al., 2015b; Zhu & Gupta, 2017; You et al., 2019), which present a more economical option than IMP, to identify winning tickets in the context of transfer learning. Experimental results demonstrate that, in contrast to the outcomes observed with randomly initialized networks (Frankle & Carbin, 2018), gradual magnitude pruning (GMP) (Zhu & Gupta, 2017) can obtain competitive subnetworks. Moreover, building upon the observation that the model update directions of a pre-trained network are more consistent than those of a randomly initialized network during training, we significantly enhance GMP's pruning-at-initialization performance by challenging conventional notions about GMP. We complement our hypothesis with extensive experiments across various models (convolutional neural networks (CNNs) (He et al., 2016), transformers (Devlin et al., 2018; Dosovitskiy et al., 2020)) in both vision and language domains and show that GMP alongside large learning rates and short pruning periods can find pruning masks comparable to or better than those of IMP for initial networks in the context of transfer learning. Finally, based on our observation that the pruning mask obtained at the end of a single training run works for LTH, we investigate whether the methodological boundary between the LTH-based and conventional pruning methods can be removed for pre-trained networks.

In summary, our **contributions** are as follows: **(i)** Our research stands as the first to systematically apply and compare diverse pruning-at-initialization methodologies to pre-trained networks, utilizing downstream datasets. Our experimental findings cast light on the limitations of saliency-based techniques in contrast to the magnitude-based baseline and IMP; **(ii)** We showcase that the achievement of a remarkable sparsity-performance trade-off is feasible when performing pruning-at-initialization on a network that is initialized with pre-trained weights for downstream tasks, without the necessity for repetitive train-prune cycles; **(iii)** Our work introduces the untapped potential of GMP within the domain of pruning-at-initialization. Our study demonstrates that GMP exhibits efficacy not only in the conventional scenario but also in the pruning-at-initialization scenario, where masks obtained at the end of the pruning process are applied to the initial weights. By challenging conventional notions about GMP, we expose its capability to significantly enhance the performance-sparsity trade-off; **(iv)** Our analysis provides empirical validation of the effectiveness of state-of-the-art pruning techniques in the realm of pruning-at-initialization, marking a pioneering contribution in this area.

## 2    RELATED WORK

Since the advent of contemporary deep learning models such as AlexNet (Krizhevsky et al., 2017), which demonstrate significant performance improvements compared to earlier versions, efforts have been initiated to incorporate pruning techniques (LeCun et al., 1990) to reduce the size of over-parameterized models. The first substantial breakthrough was achieved by implementing magnitude-

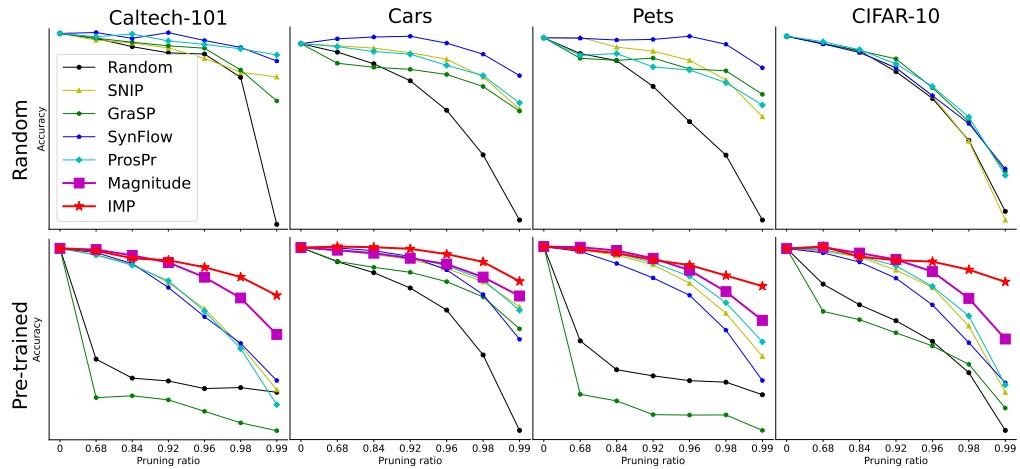

Figure 1: The results of sparse training on Caltech-101, Stanford Cars, Oxford-IIIT Pets, and CIFAR-10 for both randomly initialized (top) and ImagNet pre-trained (bottom) ResNets.

based and iterative pruning (Han et al., 2015a). This led to the evolution of an advanced method that progressively modulates the pruning rate (Zhu & Gupta, 2017). The objective of these pruning methods is to achieve a combination of fully converged weights and the corresponding pruning mask by selectively zeroing out weights during or after the training process (Zhang et al., 2022). In our paper, we will term these pruning methods "conventional pruning" for clarity.

In contrast to conventional pruning, *pruning-at-initialization* aims to find a pruning mask for a given initial network that maximizes the performance that the model can reach. IMP, a representative pruning-at-initialization method, first (i) trains the initial network until it converges; (ii) prunes a certain percentage of the weights with the smallest magnitude; (iii) returns the remaining weights to their initial values and repeats the train-prune cycles for the sparse subnetwork up to the target sparsity. IMP can successfully obtain trainable subnetworks, but it is a computation-intensive algorithm that requires many training cycles. On the other hand, the saliency-based methods measure synaptic saliency scores for given network parameters with no or a few training steps and prune based on those scores. Synaptic saliency ($\mathcal{S}$) can be formulated as $\mathcal{S} = \mathcal{R} \odot \theta_t$; where $\odot$ and $\theta$ denote the Hadamard product and network parameter at training step $t$, respectively, and $\mathcal{R}$ has different forms for each method. $\mathcal{S}$ is $\left| \frac{\partial \mathcal{L}}{\partial \theta_0} \odot \theta_0 \right|$ in SNIP (Lee et al., 2018), $-\left( H \frac{\partial \mathcal{L}}{\partial \theta_0} \right) \odot \theta_0$ in GraSP (Wang et al., 2020), $\frac{\partial \left( \vec{1}^\top \left( \prod_{l=1}^{L} |\theta^l| \right) \vec{1} \right)}{\partial \theta_0} \odot \theta_0$ in SynFlow (Tanaka et al., 2020), and $\left| \frac{\partial \mathcal{L}}{\partial \theta_T} \odot \theta_0 \right|$ in ProsPr Alizadeh et al. (2022), where $\mathcal{L}$ and $H$ denote the training loss function and Hessian matrix, respectively.

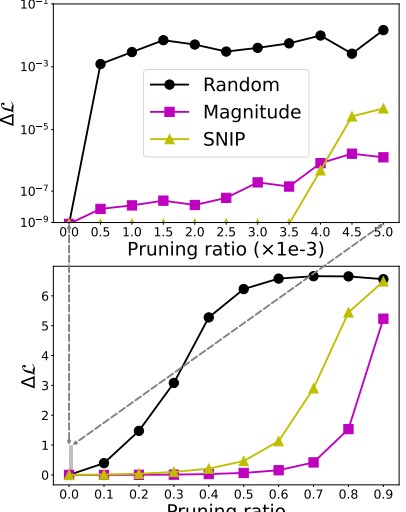

Figure 2: The average cross-entropy loss changes resulting from weight pruning the pre-trained network using the SNIP, Random, and Magnitude methods on a set of randomly selected ImageNet validation samples.

Recent studies on LTH considered pre-trained networks. Chen et al. (2021) demonstrated that they could find sparse subnetworks that maintain downstream performances for ImageNet pre-trained CNNs. They applied IMP to the pre-trained networks using the pre-training dataset. Similarly, Iofinova et al. (2022) obtained sparse subnetworks by pruning during ImageNet pre-training and proved that their downstream performances were maintained. Our study differs from theirs in that *we use downstream datasets for pruning-at-initialization*. While there is a study (Chen et al., 2020) that validated LTH for the pre-trained BERT model by applying IMP to BERT using downstream datasets, our study yields comprehensive insights that extend beyond the scope of prior observations.

## 3 PRUNING-AT-INITIALIZATION IN TRANSFER LEARNING

We observe the results of applying existing pruning-at-initialization methods to a pre-trained network. In particular, we demonstrate why the synaptic saliency-based methods fail. Figure 1 displays the results of pruning-at-initialization on Caltech-101 (Li et al., 2022), Stanford Cars (Krause et al., 2013), Oxford-IIIT Pets (Parkhi et al., 2012), and CIFAR-10 (Krizhevsky et al., 2009) for randomly initialized and ImagNet pre-trained ResNets. Here, "Random" and "Magnitude" refer to randomly selecting weights and selecting weights with the smallest magnitudes for pruning-at-initialization, respectively. For randomly initialized models (first row), the LTH-based methods exhibit effectiveness as originally claimed in their respective papers. However, we observe that for the pre-trained network (second row), those methods underperform a simple baseline, the Magnitude method. To investigate the reasons for the limitations of saliency-based methods, we examine the precision of the synaptic saliency used for pruning. Specifically, we compare the results obtained by pruning the pre-trained network using the SNIP, Random, and Magnitude methods to minimize the average cross-entropy loss changes ($\Delta\mathcal{L}$) on a set of randomly selected ImageNet validation samples. In Figure 2, when the pruning ratio is very small (top), SNIP prunes the model parameters more precisely with almost no change in loss compared to the Magnitude method. However, as the pruning ratio increases, SNIP underperforms the Magnitude method, showing that in the vicinity of the pre-trained model on the loss landscape, the loss gradient allows for a better prediction of $\Delta\mathcal{L}$, but when the pruning ratio is large and the model parameters change significantly, the nonlinearity of the loss landscape hinders the loss gradient from predicting $\Delta\mathcal{L}$. In fact, our observations align with certain papers (Frankle et al., 2020b; Liu et al., 2022) that questioned the validity of synaptic saliency. However, those studies did not provide an analysis of pruning-at-initialization for pre-trained networks akin to ours.

## 4 LIGHTWEIGHT METHODS FOR THE STABLE INITIAL NETWORK

In Figure 1, we can see that while saliency-based pruning-at-initialization methods fail for pre-trained networks, IMP can successfully find trainable subnetworks. However, IMP is a computationally intensive algorithm that requires multiple train-prune cycles. In this section, we investigate whether we can find trainable subnetworks for pre-trained networks more efficiently than IMP. Recent studies (Frankle et al., 2020a; Paul et al., 2022) have shown that trainable subnetworks can be found if and only if the subnetworks are "stable". They define stability as follows:

**Definition 1.** *(Frankle et al., 2020a) A network $\theta$ is **stable** if a pair of networks, trained from $\theta$ using the identical algorithm but with different randomness (e.g., different random seed, minibatch sampling, data augmentation, etc.), are **linearly mode connected** with high probability.*

**Definition 2.** *(Linear mode connectivity) Two networks $\theta_1$ and $\theta_2$ are **linearly mode connected** if they are connected by a linear path of intermediate networks that have interpolation errors ($\epsilon$) close to zero. $\epsilon$ can be mathematically defined as follows::*

$$\epsilon = |\lambda \cdot \mathbb{E}\left[\mathcal{L}(x, y; \theta_1)\right] + (1-\lambda) \cdot \mathbb{E}\left[\mathcal{L}(x, y; \theta_2)\right] - \mathbb{E}\left[\mathcal{L}(x, y; \lambda \cdot \theta_1 + (1-\lambda) \cdot \theta_2)\right]|, \quad (1)$$

*where $\mathcal{L}(x, y; \theta)$ denotes the loss for $\theta$ on the input-label pair (x,y), and $\lambda \in [0, 1]$.*

First, we compare the stability of randomly initialized and ImageNet pre-trained ResNets on CIFAR-100 (Krizhevsky et al., 2009) (results for other downstream datasets can be found in Appendix A). Figure 3 demonstrates that the pre-trained network is significantly more stable than randomly

---

**Algorithm 1** The tested sparse training algorithms

---

1: Initialize a network $\theta \in \mathbb{R}^d$ with pre-trained weights $\theta_0 \in \mathbb{R}^d$ and a mask $M \in \mathbb{R}^d$ with $\vec{1} \in \mathbb{R}^d$
2: **for** $r \in \{1, \ldots, R\}$ **do**   # $R$ : the number of weight resetting
3:    **for** $i \in \{1, \ldots, I\}$ **do**   # $I$ : the number of training steps
4:       Train the pruned network $\theta \odot M$
5:       **if** $i \% \pi == 0$ **then**   # $\pi$ : the interval between adjacent pruning steps (pruning period).
6:          Prune a pre-defined amount of the weights with the smallest magnitude  # update $M$
7:    Reset the trained parameters $\theta$ to the pre-trained weights $\theta_0$
8: **Evaluation of the obtained mask:** Train the sparse subnetwork $\theta_0 \odot M$

---

initialized networks. In other words, networks trained from a pre-trained network are highly likely

to converge to local minima within the same loss basin, which is consistent with previous findings (Neyshabur et al., 2020). Then, we investigate whether *the remarkable stability observed in pre-trained networks can lead to an algorithm that efficiently identifies trainable subnetworks in the context of transfer learning*. Specifically, we compare various pruning-at-initialization algorithms by applying them to transfer learning with an ImageNet pre-trained ResNet on CIFAR-100 (see Section 6 for results on other datasets). The tested pruning algorithms can be summarized as Algorithm 1.

Let $I^\star$ be the number of steps used to fully train the pre-trained model on the downstream task. Then, Algorithm 1 can be reduced to the following four pruning-at-initialization algorithms:

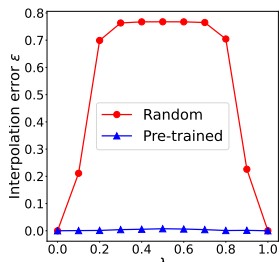

1. IMP (Frankle & Carbin, 2018): $R > 1$, $\pi = I$, and $I = I^\star$.
2. EB (You et al., 2019): $R > 1$, $\pi = I$, and $I < I^\star$.
3. One-shot (Han et al., 2015b): $R = 1$, $\pi = I$, and $I \leq I^\star$.
4. GMP (Zhu & Gupta, 2017): $R = 1$, $\pi < I$, and $I \leq I^\star$.

We use unstructured pruning commonly employed in studies on LTH. Additionally, we set the computation cost for obtaining the pruning mask of all other methods except IMP to be equal to $I^\star$ training steps. Please note that we do not use the "rewinding" strategy (Frankle et al., 2019), which involves rewinding trained weights to an earlier point in training instead of resetting them to their initial values to stabilize IMP, when applying IMP to the pre-trained model. This is because such a strategy does not produce pruning masks for initialized models, which is the main objective of pruning-at-initialization. Furthermore, a previous study (Chen et al., 2020) demonstrated that the rewinding strategy is not essential for the pre-trained network. More details are provided in Appendix C.

Figure 3: The stability of randomly initialized and ImageNet pre-trained ResNets on CIFAR-100. The test error rates are employed to measure interpolation errors.

Table 1 surprisingly demonstrates that it is feasible to identify pruning masks that are comparable to the IMP mask in terms of pruning-at-initialization while incurring significantly lower computational costs for pre-trained networks. Notably, the GMP method outperforms EB and One-shot methods for all tested pruning ratios. However, the EB, One-shot, and GMP methods exhibit noticeably inferior performance compared to IMP at the highest pruning ratio (0.99). To overcome this limitation, in the following section, we explore ways to improve the GMP method, which exhibits the closest pruning-at-initialization performance to IMP.

## 5 THE UNTAPPED POTENTIAL OF GRADUAL MAGNITUDE PRUNING

GMP (Zhu & Gupta, 2017) gradually prunes a certain percentage of network parameters based on their magnitudes at each pruning step. The gradual nature of GMP helps mitigate the sudden loss of accuracy that can occur with aggressive pruning methods. In other words, it allows for a smoother transition and provides the network with the opportunity to recover during the training phase between successive pruning steps. Therefore, it is important to appropriately set the interval between adjacent pruning steps (pruning period) and the learning rate for GMP. For instance, pruning with an excessively small pruning period enables a seamless model transition. However, it also poses challenges for the interval training steps to recover effectively from the accuracy loss incurred by zeroing out the weights. Conversely, pruning with a disproportionately high learning rate may lead to a premature pruning of weights before they fully converge to an optimal solution.

In this study, based on the consistent gradients observed during the training of pre-trained models, we propose the use of a large learning rate and a small pruning period for GMP in the context of pruning-at-initialization (the analysis regarding conventional pruning is deferred for future work). In Figure 4, we plot the Gram matrix of normalized gradients acquired during the training of randomly initialized and ImageNet pre-trained ResNets on CIFAR-100 (we provide the Gram matrix for other datasets in Appendix B); if we denote $G_{ij}$ as the element in the $i$-th row and $j$-th column of the Gram matrix, a positive value of $G_{ij}$ indicates that the $i$-th and $j$-th update directions are similar. Figure 4 demonstrates that pre-trained networks, in contrast to random initialization, exhibit smoother changes in update directions and positive correlations among adjacent update directions. Based on these observations, we posit that for pre-trained networks, employing a larger learning rate can alleviate the problem encountered during the interval training steps of GMP when using a smaller

Table 1: Performance comparison of the IMP, EB, One-shot, GMP and $\overline{\text{GMP}}$ methods on CIFAR-100 when applied to an ImageNet pre-trained ResNet. For each method, we conduct three runs and report both the average accuracy achieved and its relative cost for identifying the winning ticket, denoted as "accuracy (cost)". The best and second-best accuracies are highlighted in **bold** and underlined.

| Method | Pruning ratio | | | | | | |
|---|---|---|---|---|---|---|---|
| | 0.00 | 0.68 | 0.84 | 0.92 | 0.96 | 0.98 | 0.99 |
| IMP | | 83.94 (6) | 82.81 (9) | 82.14 (12) | 81.24 (15) | 80.47 (18) | 79.53 (21) |
| EB | 84.75 (0) | 84.06 (1) | 83.57 (1) | 82.55 (1) | 81.38 (1) | 79.55 (1) | 76.01 (1) |
| One-shot | | 83.93 (1) | 83.64 (1) | 82.39 (1) | 80.28 (1) | 75.38 (1) | 65.53 (1) |
| GMP | | **84.33** (1) | **83.91** (1) | **83.36** (1) | 81.98 (1) | 81.10 (1) | 78.66 (1) |
| $\overline{\text{GMP}}$ | 84.75 (0) | 84.13 (1) | 83.83 (1) | 83.19 (1) | **82.53** (1) | **81.49** (1) | **80.02** (1) |

Table 2: Improvements in pruning-at-initialization performance on various vision datasets resulting from the application of $\overline{\text{GMP}}$ to an ImageNet pre-trained ResNet at a pruning ratio of 0.99.

| Method | Downstream dataset | | | | |
|---|---|---|---|---|---|
| | CIFAR-10 | CIFAR-100 | Caltech-101 | Pets | Cars |
| IMP | 95.29 | 79.53 | 78.17 | 78.39 | 87.34 |
| GMP | 95.43 | 78.66 | 53.12 | 67.48 | 89.78 |
| $\overline{\text{GMP}}$ | **95.66** | **80.02** | **81.32** | **80.46** | **90.22** |

Table 3: The GMP results on CIFAR-100 with respect to different learning rates (LR) and pruning periods ($\pi$).

| LR | $\pi$ | Acc. |
|---|---|---|
| $10^{-2.5}$ | 100 | 78.66 |
| $10^{-2.5}$ | 2 | 77.76 |
| $10^{-2}$ | 2 | 80.02 |

pruning period. Specifically, while the original GMP paper (Zhu & Gupta, 2017) suggested using pruning periods between 100 and 1000, we propose using pruning periods below 100. Note that in our paper, we label the GMP with a large learning rate and a small pruning period, a departure from conventional wisdom regarding GMP usage, as $\overline{\text{GMP}}$. To demonstrate the effectiveness of $\overline{\text{GMP}}$ in terms of pruning-at-initialization, we compare the outcomes achieved through the application of $\overline{\text{GMP}}$ with the results presented in Table 1.

Table 1 demonstrates that $\overline{\text{GMP}}$ inherits the comparable or superior results achieved by GMP over IMP, while also addressing the limitations of GMP at high pruning ratios. Notably, in Table 2, we can observe substantial performance improvements brought about by employing a large learning rate and a small pruning period. For instance, regarding Caltech-101 and Oxford Pets datasets, at a pruning ratio of 0.99, GMP converges to degenerate solutions with pruning-at-initialization performances of 53.12% and 67.48%, respectively. In contrast, $\overline{\text{GMP}}$ achieves performances of 81.32% and 80.46% for each

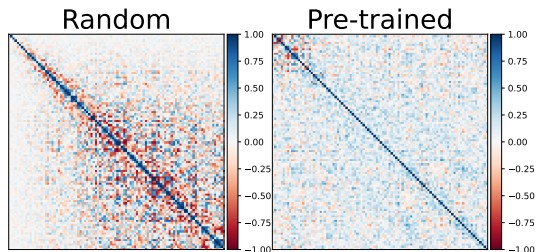

Figure 4: The Gram matrix of normalized gradients acquired during the training of randomly initialized and ImageNet pre-trained ResNets on CIFAR-100.

respective dataset at the same pruning ratio. Furthermore, Table 3 provides empirical support for our hypothesis that, for pre-trained networks, the loss incurred by using a smaller pruning period can be compensated for by employing a larger learning rate. Please be aware that, as we adapt GMP for the purpose of pruning-at-initialization, a larger learning rate is exclusively used during the generation of pruning masks for the initialized model, whereas the original learning rate is employed when training the identified sparse subnetwork for downstream tasks.

In the following sections, we validate our findings through a series of experiments conducted in diverse settings. Specifically, we assess the effectiveness of $\overline{\text{GMP}}$ when applied to an ImageNet pre-trained ResNet on various vision benchmarks. Additionally, we demonstrate the efficacy of $\overline{\text{GMP}}$ in pruning-at-initialization using vision transformers (ViTs) (Dosovitskiy et al., 2020), which have garnered significant attention in the field of computer vision. Furthermore, we establish the effectiveness of $\overline{\text{GMP}}$ not only in the vision domain but also in the language domain by conducting experiments on pre-trained transformers. Lastly, drawing upon the observation that a pruning mask

obtained at the end of a single training without weight resetting can yield favorable results for LTH, we apply a state-of-the-art pruning method (Zhang et al., 2022) to pruning-at-initialization for pre-trained networks; as a result, we reveal that the methodological distinction between the LTH-based and conventional pruning can become indistinct.

# 6 EXPERIMENTAL RESULTS AND DISCUSSION

## 6.1 EXPERIMENTAL SETUP

For vision domain experiments, we use CIFAR-10, CIFAR-100, Caltech-101, Oxford-IIIT Pets, and Stanford Cars. To assess the effectiveness of the $\overline{\text{GMP}}$ method on these vision datasets, we employ ResNet-50, ViT-B-32, and ViT-L-32 pre-trained on ImageNet. Within the natural language domain, we leverage seven datasets (CoLA, SST-2, MRPC, STS-B, QQP, MNLI, QNLI) from the GLUE benchmark (Wang et al., 2018), in addition to the SQuAD v1.1 QA dataset (Rajpurkar et al., 2016). BERT$_{\text{BASE}}$ (Devlin et al., 2018) and RoBERTa$_{\text{BASE}}$ (Liu et al., 2019) are selected as the pre-trained models for these natural language tasks. For the ImageNet pre-trained models, we explore a grid consisting of seven logarithmically spaced learning rates ranging from 0.0001 to 0.1, along with seven logarithmically spaced values of weight decay between 0.00001 and 0.01, including the option of zero weight decay. In addition, $\overline{\text{GMP}}$ demonstrates no performance degradation when the computational cost for mask generation is reduced by half compared to the original training cost ($I^\star = 20,000$ training steps). As a result, we use only half of the computational cost from the original training for the $\overline{\text{GMP}}$ method in the visual domain. The hyper-parameters for the natural language tasks are set following Chen et al. (2020). Note that all values presented in the figures and tables of this paper are the average results over three runs. Further experimental details are provided in Appendix C.

## 6.2 THE EFFECTIVENESS OF $\overline{\text{GMP}}$ ON CNNS IN THE VISION DOMAIN

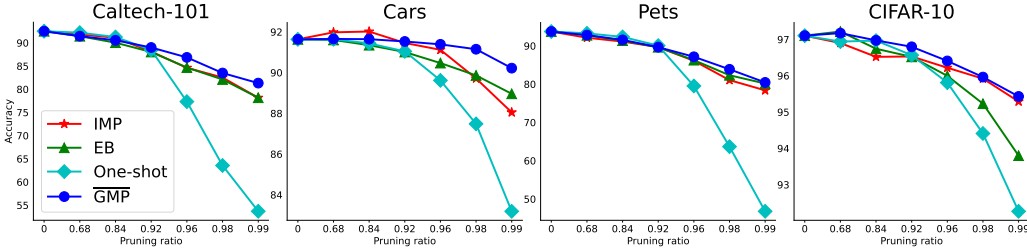

Figure 5: Comparison results of $\overline{\text{GMP}}$ with EB, One-shot, and IMP on various vision datasets in terms of pruning-at-initialization when applied to the ImageNet pre-trained ResNet-50.

In Figure 5, we present the comparison results of $\overline{\text{GMP}}$ with other methods in terms of LTH when applied to ImageNet pre-trained ResNet-50. For the ResNet-50 model, we prune all convolution layers. The results in Figure 5 align consistently with the results presented in Tables 1 and 2 for various downstream datasets. Specifically, $\overline{\text{GMP}}$ demonstrates comparable or superior outcomes compared to those of IMP, while also using much less computational costs than IMP. Additionally, it is observed that EB occasionally outperforms IMP, which can be attributed to the use of larger learning rates in EB (EB compensates for the loss arising from smaller training steps $I$ by employing larger learning rates). However, an in-depth analysis of the benefits of using larger learning rates for other pruning-at-initialization methods within the context of transfer learning falls outside the scope of our study. Thus, we defer further analysis in this direction to future research.

## 6.3 THE EFFECTIVENESS OF $\overline{\text{GMP}}$ ON TRANSFORMERS IN THE VISION DOMAIN

In this section, we evaluate the effectiveness of the $\overline{\text{GMP}}$ method on ViTs. Specifically, we prune all attention heads and MLP layers of the ViT models. Figure 6 displays the results of pruning-at-initialization for ImageNet pre-trained ViT-B-32 at a pruning ratio of 0.8. The findings exhibit similar trends to those observed for CNNs. In other words, when searching for trainable subnetworks for pre-trained ViT models, $\overline{\text{GMP}}$ achieves comparable or superior performance compared to IMP, while

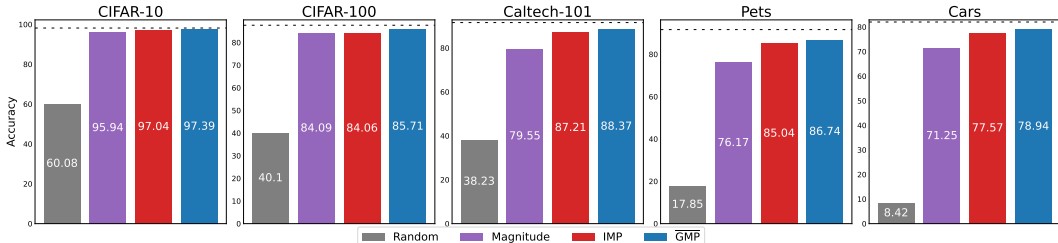

Figure 6: The results of pruning-at-initialization for ImageNet pre-trained ViT-B-32 at a pruning ratio of 0.8. The dashed line in each plot represents the performance of the original (unpruned) model.

demanding significantly lower computational costs. We provide the results of pruning-at-initialization for ImageNet pre-trained ViT-L-32 in Appendix E.

## 6.4 THE EFFECTIVENESS OF $\overline{\text{GMP}}$ ON TRANSFORMERS IN THE LANGUAGE DOMAIN

Table 4: Pruning-at-initialization result on four datasets from the GLUE benchmark and the SQuAD v1.1 dataset using pre-trained BERT model, with the best results highlighted in **bold**. The experimental results on SST-2, STS-B, and MRPC can be found in Appendix D.

| Dataset | Metric | Method | Pruning ratio | | | | |
| --- | --- | --- | --- | --- | --- | --- | --- |
| | | | 0.0 | 0.2 | 0.4 | 0.6 | 0.8 |
| MNLI | Accuracy | Random | 84.32 | 81.98 | 78.42 | 71.41 | 61.46 |
| | | Magnitude | | 83.98 | 83.71 | 82.24 | 74.22 |
| | | IMP | | 83.80 | 82.90 | 82.46 | 79.00 |
| | | $\overline{\text{GMP}}$ | | **84.60** | **84.10** | **83.54** | **81.52** |
| QQP | Accuracy | Random | 90.99 | 90.52 | 88.28 | 83.70 | 77.04 |
| | | Magnitude | | 90.83 | 90.52 | 89.61 | 85.83 |
| | | IMP | | 90.84 | 90.32 | 90.13 | 89.51 |
| | | $\overline{\text{GMP}}$ | | **91.19** | **91.09** | **90.89** | **90.30** |
| QNLI | Accuracy | Random | 91.50 | 89.31 | 83.58 | 62.80 | 60.79 |
| | | Magnitude | | 91.21 | 90.89 | 88.83 | 76.61 |
| | | IMP | | 91.30 | 90.67 | 89.66 | 86.73 |
| | | $\overline{\text{GMP}}$ | | **91.55** | **91.20** | **90.53** | **88.37** |
| STS-B | Accuracy | Random | 89.00 | 87.34 | 60.20 | 20.24 | 10.45 |
| | | Magnitude | | **88.87** | **88.51** | 86.64 | 45.12 |
| | | IMP | | 88.65 | 88.18 | **88.03** | 83.65 |
| | | $\overline{\text{GMP}}$ | | 88.50 | 87.88 | 87.23 | **84.68** |
| SQuAD | F1 score | Random | 88.65 | 85.82 | 77.24 | 26.41 | 11.72 |
| | | Magnitude | | 88.19 | 87.38 | 84.98 | 32.24 |
| | | IMP | | 88.73 | 88.07 | 87.04 | 82.22 |
| | | $\overline{\text{GMP}}$ | | **88.81** | **88.62** | **87.44** | **84.00** |

In this section, we provide evidence of the effectiveness of $\overline{\text{GMP}}$ not only in the visual domain but also in the language domain. To demonstrate this, we prune all linear layers within the transformer encoder, including the attention heads. Table 4 presents the results of pruning-at-initialization on four datasets (STS-B, QQP, MNLI, QNLI) from the GLUE benchmark, along with the SQuAD v1.1 question-answering dataset, using the pre-trained BERT model (the results on SST-2, CoLA, and MRPC, and those obtained using the pre-trained RoBERTa model can be found in Appendices D and E). The table includes the corresponding evaluation metrics for each dataset. Notably, we observe the exceptional pruning-at-initialization performance of $\overline{\text{GMP}}$ for pre-trained networks not only in the visual domain, as previously mentioned, but also in the field of natural language. Specifically, $\overline{\text{GMP}}$ consistently demonstrates superior performance across nearly all datasets and pruning ratios. However, there are instances where it exhibits slightly poorer performance compared to the Magnitude

method, particularly on relatively small datasets. For example, at a pruning ratio of 0.4, the accuracy of $\overline{\text{GMP}}$ for STS-B lags that of the Magnitude method by 0.63 percentage points. We attribute these results to the limitation of the criterion used by $\overline{\text{GMP}}$ to select weights to prune, and we investigate whether improvements in this criterion lead to improved pruning-at-initialization performance.

## 6.5 REPURPOSING A CUTTING-EDGE PRUNING METHOD TOWARDS LTH

Thus far, our observations in the context of transfer learning have revealed that it is possible to identify unimportant weights from the perspective of the initialized network for the purpose of discovering trainable subnetworks, without the need for iterative weight resetting. Remarkably, a single training session alone allows us to identify these insignificant weights selectively. Based on these motivating findings, we provide evidence supporting the feasibility of seamless integration between pruning-at-initialization and conventional pruning, particularly in the context of transfer learning. Specifically, we are the first to demonstrate that a cutting-edge pruning method can also achieve state-of-the-art performance in the LTH domain.

PLATON (Zhang et al., 2022) is a recently proposed pruning method. The authors addressed the issue of increasing uncertainty in the score used as a criterion for selecting weights to prune, which stems from the unstable training dynamics of networks. They observed significant fluctuations in these scores during training and proposed a pruning algo-

Table 5: Performance improvements in pruning-at-initialization achieved by leveraging PLATON, applied to the pre-trained BERT model. Better results are indicated in **bold**. The full version of this table can be found in Appendix D.

| Dataset | Method | Pruning ratio | | | |
|---------|--------|------|------|------|------|
| | | 0.2 | 0.4 | 0.6 | 0.8 |
| MNLI | $\overline{\text{GMP}}$ | 84.60 | 84.10 | 83.54 | 81.52 |
| | $\overline{\text{PLATON}}$ | **84.72** | **84.47** | **84.44** | **83.15** |
| QQP | $\overline{\text{GMP}}$ | 91.19 | 91.09 | 90.89 | 90.30 |
| | $\overline{\text{PLATON}}$ | **91.29** | **91.16** | **91.12** | **90.57** |
| QNLI | $\overline{\text{GMP}}$ | **91.55** | 91.20 | 90.53 | 88.37 |
| | $\overline{\text{PLATON}}$ | 91.52 | **91.70** | **90.91** | **89.75** |
| CoLA | $\overline{\text{GMP}}$ | **58.11** | 54.88 | 50.12 | 35.07 |
| | $\overline{\text{PLATON}}$ | 57.07 | **56.66** | **56.09** | **48.93** |
| STS-B | $\overline{\text{GMP}}$ | 88.50 | 87.88 | 87.23 | 84.68 |
| | $\overline{\text{PLATON}}$ | **89.36** | **89.30** | **89.08** | **87.75** |
| MRPC | $\overline{\text{GMP}}$ | **86.03** | 85.05 | 82.43 | 74.59 |
| | $\overline{\text{PLATON}}$ | 85.38 | **86.76** | **85.54** | **83.82** |

rithm that considers the uncertainty of these scores and employs exponential moving average to mitigate the instability of the scores. We denote PLATON incorporating a short pruning period and a large learning rate similar to $\overline{\text{GMP}}$, as $\overline{\text{PLATON}}$. Table 5 presents the results of pruning-at-initialization when applying $\overline{\text{PLATON}}$ to the pre-trained BERT model across five datasets from the GLUE benchmark. Remarkably, we observe that $\overline{\text{PLATON}}$ surpasses $\overline{\text{GMP}}$ by a substantial margin. Of particular significance is the conspicuous performance of $\overline{\text{PLATON}}$ on small datasets, demonstrating the transferability of the characteristics of PLATON witnessed in the original PLATON study (Table 1 in Zhang et al. (2022)) to the realm of pruning-at-initialization.

## 7 CONCLUSION

Our study is centered around the goal of exploring LTH or pruning-at-initialization on a network that is initialized with pre-trained weights, as outlined in Chen et al. (2020) and Chen et al. (2021), utilizing the provided downstream dataset. We present novel discoveries not documented in previous research, as follows: (i) Various pruning-at-initialization methods, with the exception of IMP, yield suboptimal solutions when applied to a pre-trained model using a downstream dataset in comparison to magnitude-based pruning; (ii) The resource-intensive IMP algorithm is not a necessity to obtain trainable subnetworks in the context of transfer learning; (iii) Pruning-at-initialization through GMP, using a downstream dataset, demonstrates an impressive trade-off between transfer learning performance and sparsity. Specifically, deviating substantially from the conventional usage levels of GMP, the application of large learning rates and short pruning periods significantly bolsters GMP's performance in pruning-at-initialization, particularly for high pruning ratios; (iv) In contrast to randomly initialized models, pre-trained models do not require repeated train-prune cycles for prune-at-initialization, enabling the direct application of conventional pruning methods for performance improvement in the LTH context. In Appendix F, we address the limitations of our study.

## 8 REPRODUCIBILITY

We provide our code in the supplementary material. The code enables the reproduction of the results presented in Figures 1, 5, 6, and 9, as well as the results in Tables 1, 2, 3, 4, 5, 6, and 7.

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

# Appendices

## A   STABILITY COMPARISON OF RANDOMLY INITIALIZED AND IMAGENET PRE-TRAINED RESNETS

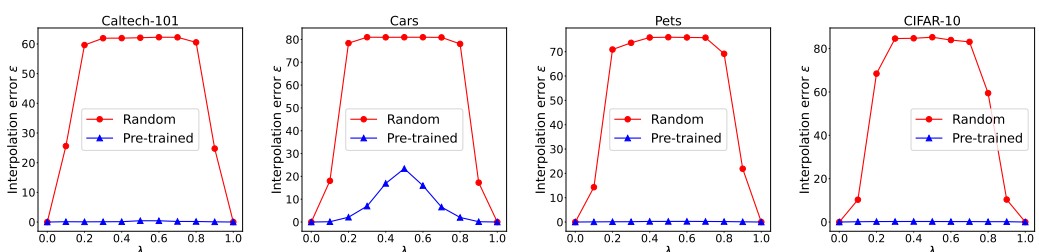

Figure 7: Stability comparisons of randomly initialized and ImageNet pre-trained ResNets on Caltech-101, Stanford Cars, Oxford-IIIT Pets, and CIFAR-10. We can see that the ImageNet pre-trained ResNet is more significantly stable than randomly initialized ResNets with respect to all tested vision benchmarks.

## B   THE GRAM MATRIX FOR OTHER DATASETS

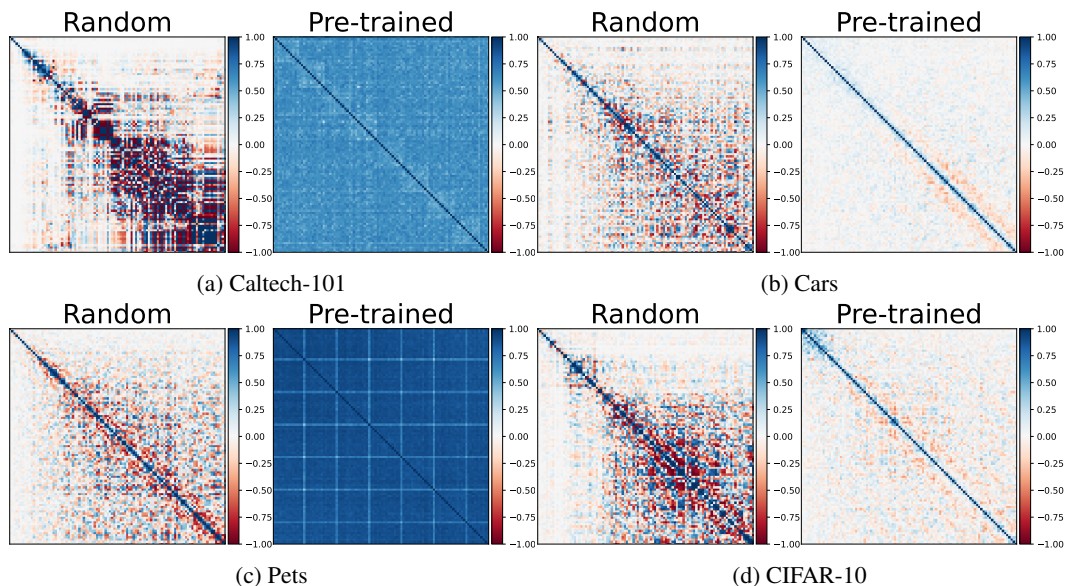

Figure 8: The Gram matrices of normalized gradients acquired during the training of randomly initialized and ImageNet pre-trained ResNets on Caltech-101, Stanford Cars, Oxford-IIIT Pets, and CIFAR-10.

Figure 8 demonstrates that pre-trained initialization, in contrast to random initialization, exhibits smoother changes in update directions and positive correlations among adjacent update directions on Caltech-101, Stanford Cars, Oxford-IIIT Pets, and CIFAR-10. In order to visually contrast the positive and negative correlations in the figures, the hyperbolic tangent function was applied.

## C  EXPERIMENTAL DETAILS

**Datasets and architectures.**  For vision domain experiments, we used the following downstream datasets: CIFAR-10, CIFAR-100, Caltech-101, Oxford-IIIT Pets, and Stanford Cars. To assess the effectiveness of the $\overline{\text{GMP}}$ method on these vision datasets, we employed ResNet-50, ViT-B-32, and ViT-L-32 pre-trained on the ImageNet dataset. Within the natural language domain, we leveraged seven datasets (CoLA, SST-2, MRPC, STS-B, QQP, MNLI, QNLI) from the GLUE benchmark (Wang et al., 2018), in addition to the SQuAD v1.1 QA dataset (Rajpurkar et al., 2016). $\text{BERT}_{\text{BASE}}$ (Devlin et al., 2018) and $\text{RoBERTa}_{\text{BASE}}$ (Liu et al., 2019) were specifically selected as the pre-trained models for these natural language tasks.

**Implementation details.**  For the ImageNet pre-trained models, we trained for 20,000 steps ($I^\star$), employing a batch size of 256 and using stochastic gradient descent with a momentum parameter of 0.9. The learning rate and weight decay were carefully chosen by performing a grid search. Specifically, we explored a grid consisting of seven logarithmically spaced learning rates ranging from 0.0001 to 0.1, along with seven logarithmically spaced values of weight decay between 0.00001 and 0.01, including the option of zero weight decay. The hyper-parameters for the natural language tasks were set following Chen et al. (2020) and Liu et al. (2019). In our experiments, we used unstructured pruning, specifically targeting different layers in different models. For the ResNet-50 model, we pruned all convolution layers. For the ViT models, we pruned all attention heads and MLP layers. Furthermore, in the case of BERT and RoBERTa, we pruned all linear layers, including the attention heads of the encoder. In the IMP method, we pruned 20% of the remaining weights at the end of each training cycle (Frankle & Carbin, 2018; Chen et al., 2021). On the other hand, the GMP method employed a cubic schedule (Zhu & Gupta, 2017; Zhang et al., 2022) for pruning. We conducted cross-validation to determine the optimal hyperparameters for $\overline{\text{GMP}}$. For instance, in our experiments with vision datasets, we executed GMP using multiple hyperparameter sets, limiting the training to only 2% to 5% of the full training (400 to 1000 steps), and chose the set that delivered satisfactory performance. This demonstrates that even when accounting for such a cross-validation process, GMP still incurs significantly lower computational costs compared to IMP. The identified hyperparameters are integrated into the provided code. In the visual domain, $\overline{\text{GMP}}$ demonstrated no performance degradation when the computational cost for mask generation was reduced by half (10,000 training steps) compared to the original training cost (20,000 training steps). As a result, in Section 6, we used only half of the computational cost from the original training for the $\overline{\text{GMP}}$ method in the visual domain.

## D  THE FULL VERSION OF TABLE 4 AND 5

Table 6 presents the results for the pre-trained BERT model, which include those omitted from Tables 4 and 5 due to space limitations.

## E  THE RESULTS OF PRUNING-AT-INITIALIZATION FOR OTHER PRE-TRAINED MODELS

Figure 9 and Table 7 show the results of pruning-at-initialization for ImageNet pre-trained ViT-L-32 and pre-trained RoBERTa model, respectively. The findings exhibit similar trends to those observed in Section 6. In other words, when searching for trainable subnetworks for pre-trained initialization, $\overline{\text{GMP}}$ achieves comparable or superior performance compared to IMP, despite requiring significantly lower computational costs. Note that, for ViT-L-32, a pruning ratio of 0.8 did not adequately illustrate the performance disparity among the compared methods. Consequently, we employed a pruning ratio of 0.9 for the results presented in Figure 9.

Table 6: Pruning-at-initialization result on seven datasets from the GLUE benchmark and the SQuAD v1.1 dataset using pre-trained BERT model, with the best results highlighted in **bold**.

| Dataset | Metric | Method | 0.0 | 0.2 | 0.4 | 0.6 | 0.8 |
|---|---|---|---|---|---|---|---|
| | | | | | Pruning ratio | | |
| MNLI | Accuracy | Random | | 81.98 | 78.42 | 71.41 | 61.46 |
| | | Magnitude | | 83.98 | 83.71 | 82.24 | 74.22 |
| | | IMP | 84.32 | 83.80 | 82.90 | 82.46 | 79.00 |
| | | GMP | | 84.60 | 84.10 | 83.54 | 81.52 |
| | | PLATON | | **84.72** | **84.47** | **84.44** | **83.15** |
| QQP | Accuracy | Random | | 90.52 | 88.28 | 83.70 | 77.04 |
| | | Magnitude | | 90.83 | 90.52 | 89.61 | 85.83 |
| | | IMP | 90.99 | 90.84 | 90.32 | 90.13 | 89.51 |
| | | GMP | | 91.19 | 91.09 | 90.89 | 90.30 |
| | | PLATON | | **91.29** | **91.16** | **91.12** | **90.57** |
| QNLI | Accuracy | Random | | 89.31 | 83.58 | 62.80 | 60.79 |
| | | Magnitude | | 91.21 | 90.89 | 88.83 | 76.61 |
| | | IMP | 91.50 | 91.30 | 90.67 | 89.66 | 86.73 |
| | | GMP | | **91.55** | 91.20 | 90.53 | 88.37 |
| | | PLATON | | 91.52 | **91.70** | **90.91** | **89.75** |
| SST-2 | Spearman correlation | Random | | 91.13 | 87.27 | 83.10 | 81.84 |
| | | Magnitude | | **92.74** | 92.24 | 91.17 | 84.06 |
| | | IMP | 92.70 | 92.66 | 92.39 | 91.17 | 88.11 |
| | | GMP | | 92.66 | 92.43 | 92.20 | 89.53 |
| | | PLATON | | **92.74** | **92.66** | **92.39** | **90.44** |
| CoLA | Matthews correlation coefficient | Random | | 41.02 | 13.49 | 08.59 | 00.00 |
| | | Magnitude | | 56.88 | 54.32 | 42.56 | 03.87 |
| | | IMP | 57.28 | 56.84 | 53.51 | 51.60 | 34.19 |
| | | GMP | | **58.11** | 54.88 | 50.12 | 35.07 |
| | | PLATON | | 57.07 | **56.66** | **56.09** | **48.93** |
| STS-B | Accuracy | Random | | 87.34 | 60.20 | 20.24 | 10.45 |
| | | Magnitude | | 88.87 | 88.51 | 86.64 | 45.12 |
| | | IMP | 89.00 | 88.65 | 88.18 | 88.03 | 83.65 |
| | | GMP | | 88.50 | 87.88 | 87.23 | 84.68 |
| | | PLATON | | **89.36** | **89.30** | **89.08** | **87.75** |
| MRPC | Accuracy | Random | | 73.69 | 70.26 | 69.04 | 68.87 |
| | | Magnitude | | 85.84 | 81.13 | 74.43 | 69.61 |
| | | IMP | 85.29 | 84.89 | 85.13 | 82.76 | 72.30 |
| | | GMP | | **86.03** | 85.05 | 82.43 | 74.59 |
| | | PLATON | | 85.38 | **86.76** | **85.54** | **83.82** |
| SQuAD | F1 score | Random | | 85.82 | 77.24 | 26.41 | 11.72 |
| | | Magnitude | | 88.19 | 87.38 | 84.98 | 32.24 |
| | | IMP | 88.65 | 88.73 | 88.07 | 87.04 | 82.22 |
| | | GMP | | **88.81** | **88.62** | **87.44** | **84.00** |

Table 7: Pruning-at-initialization results on seven datasets from the GLUE benchmark and the SQuAD v1.1 dataset using pre-trained RoBERTa model, with the best results highlighted in **bold**.

| Dataset | Metric | Method | 0.0 | 0.2 | Pruning ratio 0.4 | 0.6 | 0.8 |
|---|---|---|---|---|---|---|---|
| MNLI | Accuracy | Random | | 84.01 | 77.76 | 73.04 | 61.19 |
| | | Magnitude | 87.43 | **87.52** | **86.83** | 80.36 | 69.72 |
| | | IMP | | 87.05 | 86.33 | 85.36 | 81.48 |
| | | GMP | | 87.04 | 86.66 | **85.99** | **83.69** |
| QQP | Accuracy | Random | | 90.38 | 88.36 | 84.36 | 79.18 |
| | | Magnitude | 91.74 | 91.58 | 91.09 | 89.14 | 84.15 |
| | | IMP | | 91.67 | 91.36 | **90.77** | 89.57 |
| | | GMP | | **91.79** | **91.52** | 90.60 | **90.37** |
| QNLI | Accuracy | Random | | 87.72 | 82.88 | 65.87 | 61.49 |
| | | Magnitude | 92.72 | **92.59** | **91.48** | 85.51 | 73.09 |
| | | IMP | | 92.58 | 91.92 | 90.82 | 86.29 |
| | | GMP | | 92.06 | 91.23 | **90.98** | **88.02** |
| SST-2 | Spearman correlation | Random | | 91.06 | 86.58 | 83.49 | 83.22 |
| | | Magnitude | 94.84 | 94.30 | 93.35 | 88.57 | 82.30 |
| | | IMP | | **94.76** | **94.38** | **93.54** | 87.04 |
| | | GMP | | 94.11 | 94.00 | 92.89 | **89.64** |
| CoLA | Matthews correlation coefficient | Random | | 27.49 | 13.17 | 10.17 | 00.00 |
| | | Magnitude | 60.74 | 58.63 | 43.52 | 13.68 | 00.00 |
| | | IMP | | 60.59 | 58.97 | 44.57 | 12.36 |
| | | GMP | | **61.11** | **62.32** | **53.03** | **15.58** |
| STS-B | Accuracy | Random | | 83.23 | 55.67 | 24.84 | 19.07 |
| | | Magnitude | 90.62 | 90.33 | 89.11 | 81.43 | 41.27 |
| | | IMP | | **90.56** | **89.73** | 87.24 | 77.64 |
| | | GMP | | 90.45 | 89.63 | **88.71** | **83.98** |
| MRPC | Accuracy | Random | | 80.60 | 74.92 | 76.44 | 67.16 |
| | | Magnitude | 89.05 | 87.99 | 86.27 | 80.00 | 71.92 |
| | | IMP | | **89.79** | **87.66** | 81.29 | 65.11 |
| | | GMP | | 88.56 | 86.85 | **82.52** | **71.57** |
| SQuAD | F1 score | Random | | 87.52 | 73.94 | 26.44 | 14.61 |
| | | Magnitude | 92.05 | 92.05 | 91.12 | 87.04 | 23.18 |
| | | IMP | | 92.25 | 91.54 | **89.74** | 81.82 |
| | | GMP | | **92.35** | **91.56** | 89.53 | **83.92** |

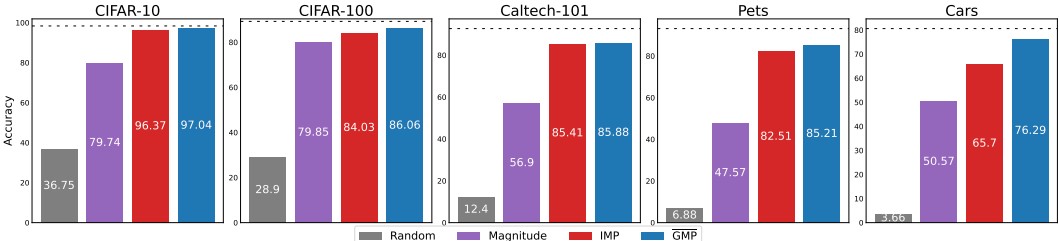

Figure 9: The results of pruning-at-initialization for ImageNet pre-trained ViT-L-32 at a pruning ratio of 0.9. The dashed line in each plot represents the performance of the original (unpruned) model.

## F LIMITATIONS

Transfer learning employs a small learning rate during downstream task learning to mitigate knowledge loss from pre-training. However, from the perspective of GMP, the use of such a small learning rate hinders smooth transitions to sparse networks due to the slow recovery between consecutive pruning steps. In other words, as evidenced in Table 3, finding the optimal learning rate and pruning period is crucial for attaining satisfactory pruning-at-initialization outcomes when implementing the $\overline{\text{GMP}}$ method. While our experiments identified the optimal hyperparameters through cross-validation, this approach limits the full realization of the benefits offered by $\overline{\text{GMP}}$. An alternative approach that adjusts hyperparameters based on training statistics between adjacent pruning steps holds promise in resolving this issue. We leave further investigation in this direction for future work.

