# OpenReview forum: "Revisiting the Lottery Ticket Hypothesis for Pre-trained Networks"
_ICLR.cc/2024/Conference — ICLR 2024 Conference Withdrawn Submission_

### Official Review · Reviewer_V5JA · 2023-10-31

**Soundness:** 3 good
**Presentation:** 3 good
**Contribution:** 2 fair
**Rating:** 5
**Confidence:** 3

**Summary:**

This paper tackles the interesting and practically useful problem of the lottery ticket hypothesis in the transfer learning setting. The paper first demonstrates that the multiple pruning-at-initialization techniques are likely to find the worst subnetwork (winning ticket)  compared to the iterative magnitude pruning (IMP) in the pre-trained network. By realizing the fact that despite having the effective performance of IMP it remains computationally expensive, the paper then uses the idea of gradual magnitude pruning (GMP) that provides comparable/better performance as that of the IMP while being computationally cheap. The experimentation conducted on multiple architectures with multiple tasks (language as well as vision) justifies the effectiveness of GMP and its variants.

**Strengths:**

* The paper has tackled an interesting and useful problem of the lottery ticket hypothesis in the transfer learning setting. The finding could be extremely useful to come up with the sparse variants of Large Language Models (LLMs).
* The paper has empirically shown an interesting phenomenon that the majority of the pruning-at-initialization techniques are likely to suffer in the pre-trained network setting. This phenomenon is shown clearly with the help of multiple figures and also intuitive justification is provided for that.
* The extensive evaluation is conducted by considering multiple architectures (ResNet and Transformers) along with multiple datasets consisting of both vision as well as language tasks. This helps to strengthen the credibility of the findings presented in the paper.

**Weaknesses:**

* The proposed technique has a limited novelty. Most of the findings are empirical-based without deeper theoretical underpinning. The paper has made a trivial extension of the GMP model which may not be enough in terms of novelty.
* The authors may need to provide more descriptions of IMP, EB, One-Shot, and GMP techniques in order to make the paper self-contained.
* The main advantage of using GMP over IMP is in terms of computational cost as the performance is comparable in most cases. The author may need to include a separate subsection in the experimentation section to compare the computational costs of different baselines along with IMP and its variants.
* The authors have mentioned in the paper that using GMP in the pre-trained network, a smaller pruning period, and a higher learning rate are preferred in order to achieve a better performance. A more comprehensive study may be required to support the above claim. Specifically, in Table 2 it would be interesting to see the performance variation on a wide range of learning rates along with the different pruning periods.

**Questions:**

Please see weaknesses section

---

### Official Review · Reviewer_mi9v · 2023-10-31

**Soundness:** 1 poor
**Presentation:** 2 fair
**Contribution:** 1 poor
**Rating:** 3
**Confidence:** 4

**Summary:**

The paper revisits LTH in the context of transfer learning and claims to unveil some novel insights such as:

1. pruning-at-initialization methods are likely to find worse pruning masks than a simple magnitude-based pruning method for pre-trained network.
2. The resource-intensive IMP algorithm is not a necessity to obtain trainable subnetworks in the context of transfer learning;
3. Pruning-at-initialization through GMP, using a downstream dataset, demonstrates an impressive trade-off between transfer learning
performance and sparsity

**Strengths:**

Strengths:

1. A good amount of experiments have been done to support the arguments made in the paper.
2.

**Weaknesses:**

I have many significant concerns with the novelty and necessity of this work. There are several claims which are over-sold in the paper e.g. 1. we are the first to demonstrate that a cutting-edge pruning method can also achieve state-of-the-art performance in the LTH domain, 2. first to systematically apply and compare diverse pruning-at-initialization methodologies to pre-trained networks, among many others. I highly recommend the authors to carefully dig up LTH/pruning for pre-trained models papers. The limited role of IMP for pre-trained models in comparison to one-shot is also established (check https://arxiv.org/abs/2306.03805). GMP vs GMP-BAR with difference of large learning rate and a small pruning period - I do not understand why this is important and shown as one of the main contributions. Isn't these are hyperparameters that can be tuned depending on intuitive performance? The addition of some large-scale models like LLMs can bring some new insights PaI which I think is not sufficiently explored. Also, the paper writing can be significantly improved, making smaller sentences to improve readability and convey a clear message.

**Questions:**

See above.

---

### Official Review · Reviewer_kmMF · 2023-10-31

**Soundness:** 2 fair
**Presentation:** 3 good
**Contribution:** 2 fair
**Rating:** 3
**Confidence:** 5

**Summary:**

This paper aims to identify sparse trainable weights (winning tickets) of a pretrained model for downstream datasets, i.e., in a transfer learning setting. They arm an existing pruning method, GMP (gradual magnitude pruning), with a larger LR and find this can significantly improve the performance. This gives them the freedom to reduce the pruning interval so as to shorten the total epochs, thus making the final model perform comparably (or better) than the other popular method IMP (iterative magnitude pruning), but using much less cost. On CNNs and transformers, the results support their findings.

**Strengths:**

1. The paper studies LTH in the transfer learning setting, which is relatively less under-explored in the community.
2. Empirically, the method can find sparse models more efficiently than the counterparts, with less training cost.
3. The paper presents quite diverse results, on both vision and nlp domains, to show the findings are generalizable.

**Weaknesses:**

I have major concerns regarding its motivation and the real contribution.

1. I am rather confused by the settings of this work - The paper states "Our research stands as the first to systematically apply and compare diverse **pruning-at-initialization** methodologies to **pre-trained** networks." Pruning-at-initialization means the model is supposed to be randomly initialized, however, the paper says the method is applied to a pretrained network. Then why call it "pruning at initialization"?

2. Limited true technical contribution.

2.1 The method, GMP_bar, is basically identical to GMP, just using a large LR. I am not sure this can be claimed as a "new" method...

2.2 Meanwhile, the observed effect that using a larger LR can boost pruning performance in Tab. 2 & 3 is not new, either -- Many works have observed this phenomenon (see the discussions in https://arxiv.org/abs/2301.05219).

Most of the performance improvements go back to the usage of a larger LR, even for the counterpart method ("*it is observed that EB occasionally outperforms IMP, which can be attributed to the use of larger learning rates in EB*"). I do not think this can be a novel point for this paper as finetuning LR is essentially not about pruning (especially when this has been reported in the community).

In short, the real contribution of this paper, in my view, is to exploit an *already-known* phenomenon (a larger LR can help boost pruning performance) in the setup of PaI for transfer learning. To me, this seems not sufficient for an ICLR paper.

**Questions:**

What is the learning rate schedule used for IMP? How many iterative cycles are used for IMP?

---

### Official Review · Reviewer_6QxM · 2023-11-01

**Soundness:** 3 good
**Presentation:** 4 excellent
**Contribution:** 3 good
**Rating:** 5
**Confidence:** 4

**Summary:**

This study delves into the Lottery Ticket Hypothesis (LTH) within the context of transfer learning on downstream datasets. The research reveals that the application of conventional pruning-at-initialization (PAI) methods, such as SNIP, GraSP, and SynFlow, to a pretrained network results in suboptimal outcomes. Conversely, Iterative Magnitude Pruning (IMP) successfully identifies trainable subnetworks. Subsequently, the paper highlights the efficacy of gradual magnitude pruning (GMP) in achieving significant improvements in transfer learning performance and introduces $\bar{GMP}$  to further enhance results through with larger learning rate and smaller pruning period. Moreover, extensive experiments across various transfer learning tasks in both vision and language domains empirically validate the effectiveness of the proposed $\bar{GMP}$.

**Strengths:**

1. The suggested $\bar{GMP}$ effectively sustains the performance of dense networks even at higher sparsity ratios (>90\%), surpassing the performance of IMP while eliminating the laborious pruning-retraining iterations required by IMP. I firmly believe that this straightforward yet highly efficient method holds substantial value in practical scenarios, particularly in the context of transfer learning with constrained computational resources.


2.  This paper rigorously conducts extensive experiments across a variety of vision and language tasks, providing compelling evidence that underscores the effectiveness of the proposed $\bar{GMP}$.


3. This paper is in general well-written and easy to follow.

**Weaknesses:**

1. The conclusion drawn regarding PAI methods in Section 3 may be subject to potentially unfair comparisons. It's important to note that these sparse training methods are originally designed to derive improved pruning masks from random initialization, rather than from fully-trained networks. Therefore, their effectiveness should be gauged based on their ability to outperform random pruning. Notably, in Figure 1, most PAI methods, with the exception of GraSP, still exhibit superior performance compared to random pruning (depicted by the black line) in both random initialization and pre-trained weights. This suggests that these methods remain effective in the context of transfer learning. Conversely, the practice of employing magnitude-based pruning on pre-trained dense weights is often categorized as pruning after training (Han et al., 2015a). Although the pre-trained networks were not specifically trained on the downstream datasets, the pre-trained weights still encapsulate valuable information and differ significantly from randomly initialized weights. Consequently, from my personal perspective, comparing PAI methods with magnitude-based pruning may be considered an inequitable assessment, even within the framework of transfer learning. Thus, it would be inaccurate to claim that saliency-based PAI methods are ineffective for pre-trained networks.


2.  The distinction between this paper and prior works (Chen et al., 2020, Chen et al., 2021) is somewhat vague. As of now, the primary dissimilarity lies in the fact that this paper places a distinct emphasis on leveraging downstream datasets for PAI. Thus, the novelty of this paper is kind of limited.


3.  The ablation studies about the learning rate and the pruning periods are missing. Although Table 3 does provide results involving different learning rates and pruning periods, the performance within the range of pruning periods spanning from 100 to 1000 remains ambiguous. This paper only show that $\bar{GMP}$ outperforms GMP in Table 1 and Table 2. However, the underlying rationale for the choice of these specific hyperparameters requires more comprehensive elucidation.

**Questions:**

Please address the above weaknesses in the rebuttal.